# An Approach to Reduce Thermal Damages on Grinding of Bearing Steel by Controlling Cutting Fluid Temperature

Raphael Lima de Paiva [1,2] , Rodrigo de Souza Ruzzi [2,3] and Rosemar Batista da Silva [2,*]

1    School of Mechanical Engineering, Campus Univ. Min. Petronio Portella, Federal University of Piaui (UFPI), Ininga, Teresina 64000, PI , Brazil; raphaellimap@hotmail.com
2    School of Mechanical Engineering, Federal University of Uberlandia (UFU), Av. João N. de Avila, 2121, Uberlandia 38400, MG , Brazil; rodruzzi@gmail.com
3    Department of Mechanical Engineering, Inga University Center (UNINGA), Rod. PR-317, 6114, Parque Industrial 200, Maringa 87000, PR , Brazil
*    Correspondence: rosemar.silva@ufu.br

**Abstract:** The use of cutting fluid is crucial in the grinding process due to the elevated heat generated during the process which typically flows to the workpiece and can adversely affect its integrity. Considering the conventional technique for cutting fluid application in grinding (flood), its efficiency is related to certain factors such as the type of fluid, nozzle geometry/positioning, flow rate and coolant concentration. Another parameter, one which is usually neglected, is the cutting fluid temperature. Since the heat exchange between the cutting fluid and workpiece increases with the temperature difference, controlling the cutting fluid temperature before its application could improve its cooling capability. In this context, this work aimed to analyze the surface integrity of bearing steel (hardened SAE 52100 steel) after grinding with an $Al_2O_3$ grinding wheel with the cutting fluid delivered via flood technique at different temperatures: 5 °C, 10 °C, 15 °C as well as room temperature (28 $\pm$ 1 °C). The surface integrity of the workpiece was analyzed in terms of surface roughness (Ra parameter), images of the ground surface, and the microhardness and microstructure beneath the machined surface. The results show that the surface roughness values reduced with the cutting fluid temperature. Furthermore, the application of a cutting fluid at low temperatures enabled the minimization of thermal damages regarding visible grinding burns, hardness variation, and microstructure changes.

**Keywords:** grinding; cutting fluid temperature; SAE 52100 steel; surface integrity; thermal damages

## 1. Introduction

Grinding is an abrasive machining process usually applied to provide a combination of good surface finishing and close tolerance for a given component. As a finishing process, the material removal rate is generally small, and combined with high cutting speeds (>30 m/s), it results in a high specific cutting energy, typically 10 times higher than machining processes with defined geometry tools such as milling [1]. Such high specific energy is also attributed to the material removal mechanism during grinding, which includes the considerable contributions of rubbing and ploughing [2].

According to Malkin and Guo [3], virtually all the grinding input energy is converted into heat which is concentrated within the cutting zone. Since the chips generated during grinding are very small and conventional abrasive materials such as aluminum oxide ($Al_2O_3$) are poor heat conductors at high temperatures [4,5], most of the heat generated flows to the workpiece. This leads to the development of high temperatures at the cutting zone and, depending on its level and time of action, the workpiece's surface integrity can be compromised due to thermal damages, which is the major limiting factor in grinding production according to Irani et al. [6].

When grinding hardened steels, workpiece burn is the most common type of thermal damage. It occurs on a machined surface and/or beneath it, resulting in a reduction in hardness due to tempering and/or rehardening, which is due to the formation of untampered martensite [3]. In this sense, focusing on alternatives to prevent or attenuate possible thermal damages is particularly interesting considering the grinding of hardened SAE 52100 steel. This specific alloy is the main material used for bearing applications and it is therefore frequently machined in its hardened state [7,8]. The use of a turning or milling process in this case is quite challenging due to the material hardness, as well as the required finishing. Thus, the grinding process is typically the first choice for machining this material in its hardened state. Additionally, considering the high heat generation during grinding and that this particular alloy is highly susceptible to thermal damage, the cooling–lubrication condition plays an important role in the grinding process of this material [9].

According to Marinescu et al. [10], the major functions of cutting fluids in the grinding process are to provide lubrication at the interface between the abrasive grains and the workpiece and cool the cutting zone and workpiece through the absorption and transportation of the heat generated during the process. Debnath et al. [11] classified the cutting fluids generally used in conventional machining processes into three different groups: oil-based (neat oils), water-based (solutions and emulsions), and gas-based coolant–lubricants. Regarding the grinding process in particular, the water-based cutting fluids are the first choice due to their cooling capability and relatively low cost, especially when the cutting fluid is applied using the conventional technique (flood application).

Given the importance of cutting fluids in the grinding process, many works have focused on analyzing the influence of some cutting fluid characteristics/parameters on different output variables such as workpiece surface integrity, cutting forces, cutting temperature, and grinding wheel wear.

Ebbrell et al. [12] tested three different nozzle positions (angular, intermediate, and tangential) for the application of a 2% emulsion cutting fluid via conventional technique (flood) in the grinding of a mild steel with an $Al_2O_3$ grinding wheel. According to the authors, the nozzle position affected the cutting fluid quantity passing beneath the grinding wheel. The best results in terms of surface finish (low values of Ra) were found by raising the nozzle position above the area of reversed flow, which reduced the hydrodynamic effect of the grinding wheel, thereby increasing the cutting fluid's penetration efficiency at the cutting zone. Aiming to improve cutting fluid's penetration efficiency, the cooling–lubrication strategy know as Very Impressive Performance Extreme Removal (VIPER) grinding was developed by Rolls–Royce for grinding difficult-to-machine materials such as nickel-based superalloys. The principle of VIPER grinding is injecting the coolant into the grinding wheel at pressures of up to 0.7 MPa, for instance [13,14].

Tawakoli et al. [15] compared the performance of oil and emulsion in the grinding of SAE 52100 hardened steel (60 HRC) with the CBN grinding wheel. The authors observed that the use of oil resulted in lower cutting forces in comparison to emulsion, which was attributed to the better lubrication capacity of the oil, improving tribological conditions and chip removal properties. Nevertheless, the authors concluded that the results could be different if other types of oil or emulsion were chosen, thus highlighting the importance of cutting fluid type for the grinding process. Alberdi et al. [16] compared two different nozzle geometries for conventional cutting fluid application on the surface grinding of GGG70 cast iron and observed that the optimized nozzle design contributed to the improved surface integrity of the workpiece and increased wheel life by 25% in comparison to the well-known Webster-based nozzle geometry.

De Paiva et al. [17] evaluated the surface integrity (roughness and microhardness) of VP 80 mold steel (45 HRC) after grinding with cutting fluid at different concentrations (proportion of oil in water). Their results showed that the cutting fluid with a higher concentration (larger percentage of oil) generated the best results in terms of surface roughness and microhardness beneath the machined surface when grinding using more severe cutting conditions. On the other hand, the authors reported that grinding with the

cutting fluid at lower concentration provided better results when grinding under mild cutting conditions.

Hadad et al. [18] tested the use of different cutting fluids applied with a minimum quantity of lubrication (MQL) technique in comparison to the conventional technique (flood) in the grinding of SAE 52100 hardened steel (50 ± 2 HRC) and reported that not only did type of fluid influence the output parameters (surface integrity, cutting forces and temperature), but it also influenced the type of cutting fluid application. According to their findings, grinding with the MQL technique reduced the surface roughness and cutting forces due to the better lubrication capacity of this technique in comparison to the conventional one. In terms of grinding temperature, however, the MQL technique could not meet the grinding cooling requirements. Damasceno et al. [19] evaluated three different cooling–lubrication techniques (MQL, conventional flood, and optimized) during the grinding of hardened 4340 steel. The authors observed that the grinding performance was strongly affected by the cooling–lubrication technique and concluded that, between the tested techniques, the optimized one outperformed the other.

Verma et al. [20] carried out grinding experiments on hardened AISI 52100 steel (62 HRC) with an $Al_2O_3$ grinding wheel and deionized water as a cutting fluid, which was applied via MQL technique under different flow rate and air pressure conditions. Tests under dry and flood conditions were also performed for comparison. The authors observed that grinding forces and residual stress decreased with the increase in flow rate and compressed air pressure. According to them, higher flow rates and compressed air pressure increases the wettable area, thereby resulting in enhanced lubrication. This reduces grinding forces and grinding temperature as well, consequently decreasing the residual stress on ground surface.

Da Silva et al. [21] studied the surface finish (3D parameters) and residual stress of AISI 4340 hardened steel after grinding using different cutting fluids applied by conventional technique (flood). The authors tested a mix cutting fluid (50% mineral-based oil and 50% soybean oil), as well as synthetic and mineral (integral) ones. Grinding experiments using the MQL technique with vegetable-based oil were also performed. The authors observed that the type of cutting fluid influenced the surface integrity of the ground parts. In terms of surface finish (Sa parameter), the best results were found after grinding with integral oil, which was attributed to its higher lubrication capacity. The MQL technique and the synthetic fluid presented higher values of Svk in comparison to the integral and mix fluid, which increases the lubricant retention capacity of the ground surface. All ground parts exhibited compressive residual stress, regardless of the cutting fluid or application technique used.

Awale et al. [22] carried out a multi-objective optimization study of MQL parameters during the grinding of a hardened AISI H13 tool steel with the $Al_2O_3$ grinding wheel. The authors tested different conditions in terms of compressed air pressure (P—0.2, 0.3, 0.4 and 0.5 MPa), cutting fluid flow rate (Q—50, 100, 150, and 200 mL/h) and stand-off distance ($d_s$—40, 50, 60, and 70 mm). According to their findings, the MQL parameters of P = 0.4 MPa, Q = 200 mL/h, and $d_s$ = 50 mm presented the best grinding performance by reducing the grinding force, specific energy, grinding temperature, and surface roughness.

De Paiva et al. [9] analyzed the influence of the cutting fluid type and its application technique on the surface integrity of SAE 52100 hardened steel after grinding with the $Al_2O_3$ grinding wheel and using semisynthetic and synthetic fluids, applied with flood and MQL techniques. The findings showed that the combination of the type of fluid and application technique strongly affected the surface integrity of ground parts. The best results were generally observed after grinding with semisynthetic fluid due to its better lubrication capacity. Additionally, the MQL technique promoted equivalent or even better results in comparison to the flood technique, especially when using the synthetic fluid.

The solid particles dispersed in cutting fluid have been used with the aim of improving the MQL technique. Lee et al. [23] observed a reduction in the cutting forces and roughness

of the machined surface after grinding a tool steel with the MQL technique using mineral oil enhanced with diamond and $Al_2O_3$ solid particles. Li et al. [24] tested the application of the MQL technique with graphene dispersed in the cutting fluid in the grinding of titanium alloy TC4. The authors observed that, compared to the traditional MQL technique (without solid particles), the use of MQL with graphene reduced the peak temperature during grinding. In a more recent work, De Paiva et al. [25] tested the use of the MQL technique with graphene in the grinding of hardened SAE 52100 with an $Al_2O_3$ grinding wheel. Their findings show that the use of graphene dispersed in the cutting fluid reduced surface roughness (Ra parameter) by up to 12% and improved the surface morphology of the ground surface.

The above literature review shows that the common cutting fluid characteristics or parameters typically studied are the type of cutting fluid, nozzle geometry and positioning, cutting fluid concentration, application technique, flow rate and pressure, and additives. The temperature of the cutting fluid, on the other hand, is usually concerned with respect to non-conventional cutting fluids such as liquid nitrogen (cryogenic grinding), cooled compressed air, and carbon dioxide ($CO_2$). Sanchez et al. [26], for instance, proposed a combination of cooled $CO_2$ and neat oil applied with the MQL technique in the grinding of AISI D2 tool steel with an $Al_2O_3$ grinding wheel. Some other examples regarding non-conventional cooling–lubrication strategies can be found in the literature, such as the works carried out by Manimaran et al. [27], Manimaran and Kumar [28], Ben Fredj et al. [29], Lopes et al. [30], Ribeiro et al. [31], Saberi et al. [32], Choi et al. [33], and Khanna et al. [34].

Considering the fact that cutting fluids are generally applied in grinding operations (neat oils, emulsion, synthetic, and semi-synthetic fluids), the temperature of the fluid in the instant of its application is not usually discussed, especially for conventional cooling (flood technique), which is the leading method used to remove the excess heat generated in grinding [35]. The temperature is an important factor related to the cooling capability of the fluid, once the heat exchange between the fluid and the contact zone is due to forced convection, which is proportional to temperature difference. In this sense, Gao et al. [36] proposed an active cutting fluid cooling system for application in grinding that uses a common air conditioner to maintain the cutting fluid at low temperatures (<3 °C). However, no experimental trials analyzing the influence of using this system on output parameters, such as the workpiece surface integrity, have been presented by the aforementioned authors. Furthermore, no similar works discussing the influence of the temperature of conventional cutting fluids were found in the literature.

In this context, the purpose of this work was to fill this gap in the literature by evaluating the grinding performance of the SAE 52100 hardened steel using a conventional water-based cutting fluid at different temperatures: 5 °C, 10 °C, 15 °C, and 28 ± 1 °C (room temperature). The grinding performance was analyzed based on roughness, images of ground surface, and the microhardness and microstructure beneath machined surface.

## 2. Materials and Methods

The grinding tests were conducted on a peripheral surface grinder machine P36, MELLO (MELLO S.A MACHINE AND EQUIPMENT, Sao Paulo, Brazil), with 2.2 kW power and 2400 rpm of maximum spindle speed. A white aluminum oxide ($Al_2O_3$) grinding wheel with 297 mm of external diameter was employed. Its designation is 38A46K6V, which stands for white fused aluminum oxide abrasive grain (99.8% pure), with a mesh grit size of #46, medium hardness and structure, and vitrified bond. The white aluminum oxide is the most applied abrasive type in the grinding of steels in general [3]. The workpiece material was the SAE 52100 hardened steel (60 ± 2 HRC), with dimensions of 18 mm and 19 mm (diameter and height, respectively). The material chemistry of SAE 52100 is shown in Table 1.

**Table 1.** Chemical composition of SAE 52100 hardened steel [37].

|        | C    | Mn    | Cr    | Si    | S     | Fe     |
|--------|------|-------|-------|-------|-------|--------|
| (wt %) | 1.03 | 0.120 | 1.526 | 0.207 | 0.042 | 97.074 |

The cutting parameters used in grinding experiments were grinding wheel speed ($v_s$) of 37 m/s, workspeed ($v_w$) of 3 m/min and radial depth of cut ($a_e$) of 30 μm. The grinding process was performed using a cross-feed strategy, with a depth of cut ($a_p$) of 0.9 mm as schematically shown in Figure 1. All the cutting parameters were kept constant for all grinding experiments. The grinding wheel was dressed prior to each experiment by using a single-point diamond dresser with a dressing depth ($a_{ed}$) of 15 μm and a dressing speed ($v_d$) of 140 mm/min (grinding wheel overlap ratio–$U_d$ = 3).

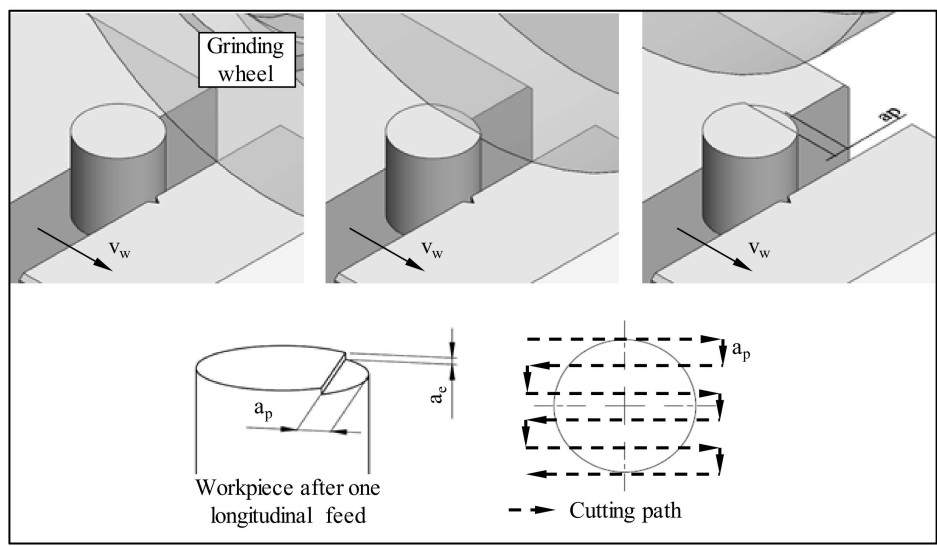

**Figure 1.** Sketch of the grinding operation used in the experimental trials.

The cutting fluid was applied via conventional technique (flood) at four different temperatures: 5 °C, 10 °C, 15 °C, and room temperature (28 ± 1 °C). The cutting fluid used was a semisynthetic vegetable-based oil, at a 5% concentration (1:20 dilution in water), and a flow rate of 9 L/min, tangentially positioned to the grinding wheel. The cutting fluid temperature was controlled in the ice bath and a thermometer was used to monitor its temperature. Once the target temperature was reached, the pump for the cutting fluid was positioned and the grinding experiment was performed. After the grinding process, the cutting fluid was returned to a different reservoir after its application in order to avoid temperature changes of the cooled cutting fluid. In Figure 2, a sketch of the grinding experiments is shown.

Regarding the experiment planning, each grinding test was replicated once. Thus, considering the variation in the cutting fluid temperature in four (4) levels, with all the other cutting parameters kept constant, a total of eight (8) grinding experiments were conducted in the present work. After each experiment, the surface integrity of the workpiece was analyzed in terms of surface roughness (Ra parameter), the appearance of ground surface, and the microhardness and microstructure beneath machined surface. Table 2 summarizes the grinding conditions used in this work, as well as the analyzed output parameters.

The workpiece's surface roughness was measured using a portable surface roughness tester SJ-201P (Mitutoyo, Kawasaki, Japan) at five (5) different regions of the machined surface, perpendicularly to the grinding direction; a filter wavelength (cut-off) of 0.8 mm and an evaluation length of 4.0 mm were used. The average standard deviation values of Ra for each grinding condition, considering both the test and replica, were considered

for analysis. The image (appearance) of the machined surface was examined after each grinding experiment by using a stereo microscope BX51 (Olympus, Tokyo, Japan).

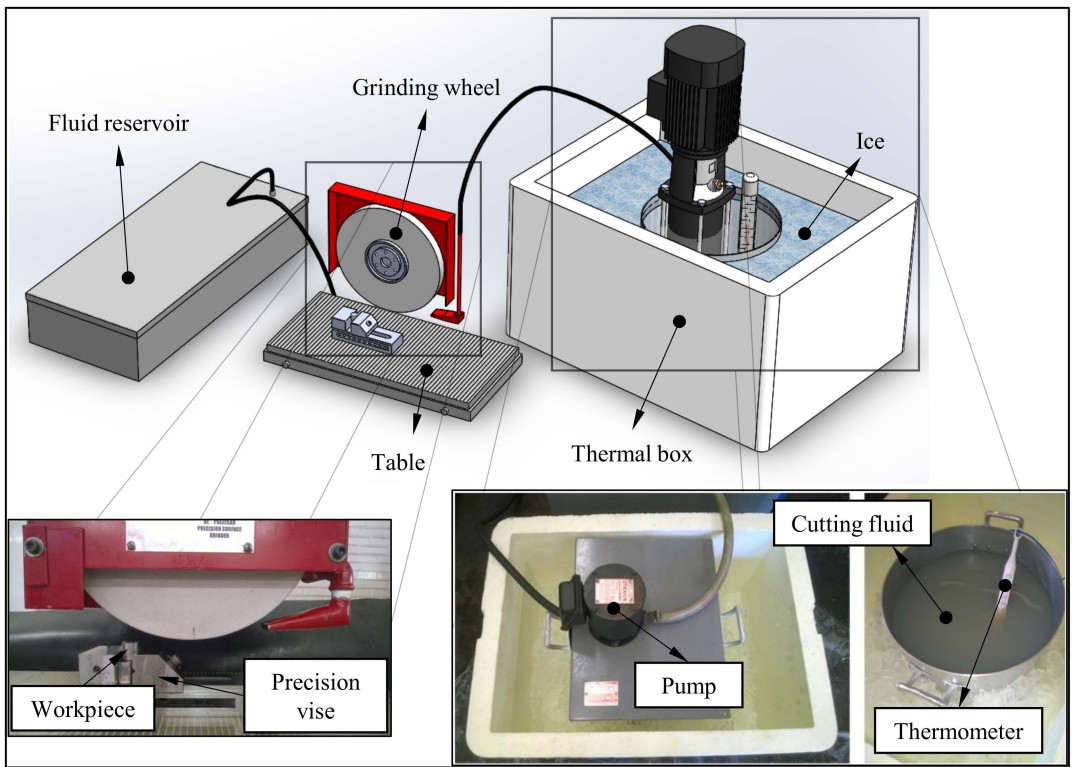

**Figure 2.** Schematic of the set-up used for grinding experiments.

**Table 2.** Grinding conditions.

| Grinding Operation | Surface Grinding |
|---|---|
| Grinding wheel | $Al_2O_3$–38A46K6V–Ø 297 mm |
| Workpiece | 18 mm × 19 mm (diameter × height)<br>SAE 52100 hardened steel–60 ± 2 HRC |
| Grinding wheel speed ($v_s$) | 37 m/s |
| Workspeed ($v_w$) | 3 m/min |
| Depth of cut per longitudinal pass ($a_p$) | 0.9 mm |
| Radial depth of cut ($a_e$) | 30 µm |
| Cutting environments | Cutting fluid at different temperatures:<br>5 °C, 10 °C, 15 °C and 28 ± 1 °C (room temperature–RT) |
| Cutting fluid | Semisynthetic vegetable-based oil<br>Concentration–1:20 (5%), flow rate–9 L/min |
| Dressing conditions | $a_{ed}$ = 15 µm<br>$v_d$ = 140 mm/min<br>$U_d$ = 3 |
| Output parameters | Surface roughness (Ra parameter)<br>Images of machined surfaces<br>Microhardness and microstructure beneath machined surface |

The microhardness was measured using an HMV Micro Hardness tester (Shimadzu, Kyoto, Japan), following the metallographic preparation of the workpieces; measurements were performed at every 20 µm beneath the machined surface until a 300 µm distance with a load of 490.3 mN (HV 0.05) applied for 10 s was achieved. The microhardness

measurements were replicated twice, and the mean values and standard deviations for each grinding condition, including the test and replica, were considered for analysis. After the microhardness measurements, the samples were polished and etched for 10 seconds with 5% Nital in order to verify the influence of the cutting fluid temperature on the workpiece microstructure after grinding.

## 3. Results and Discussion

### 3.1. Surface Roughness

The surface roughness (Ra parameter) of ground surfaces is shown in Figure 3 as a function of the cutting fluid temperature. Each Ra value corresponds to the average of ten measurements, including the test and replica. The error bars represent the standard deviation of the measurements. One notes from Figure 3 that the Ra mean values are in the range of 0.20–0.30 μm, which is in good agreement with the expected surface finish in grinding: 0.1–1.6 μm [38]. Furthermore, it can be noticed that the application of cutting fluid at low temperatures contributed to the improved surface quality of the workpiece (lower values of Ra). On average, the lowest value of the Ra parameter was 0.20 μm, observed after grinding with cutting fluid at 5 °C, which is a 31% reduction in comparison to the Ra parameter measured after grinding with the cutting fluid at room temperature (0.29 μm). Additionally, it can be noticed from Figure 3 that, considering the conditions tested in this work, there is a strong linear relationship between the mean values of Ra and the cutting fluid temperature ($R^2 > 0.9$).

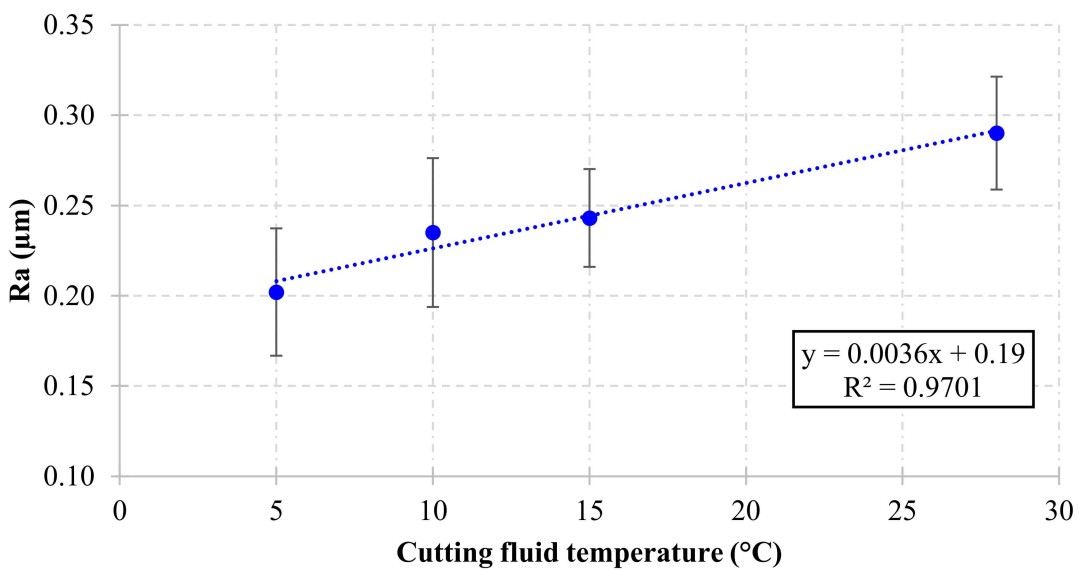

**Figure 3.** Surface roughness (Ra parameter) of the ground surface as a function of cutting fluid temperature.

The lower values of surface roughness after grinding with cutting fluid at low temperatures is associated with the material removal mechanism and thermal distortions. The softening of the workpiece material due cutting zone temperature was attenuated when the cutting fluid was applied at low temperatures, contributing to the micro-cutting mechanism instead of micro-ploughing during chip formation. Furthermore, the more efficient heat dissipation promoted by the cutting fluid at low temperatures reduces workpiece distortion during the process. Both these factors contribute to improving the surface finish (lower values of Ra), which is similar to what is usually reported in the literature regarding cryogenic grinding [27].

Manimaran and Kumar [28] analyzed the surface finish of AISI 316 stainless steel after grinding with an $Al_2O_3$ grinding wheel and liquid nitrogen as a cutting fluid. As a result, they reported that the cryogenic grinding provided a reduction in surface roughness (Ra parameter) of 26% compared to conventional cutting fluid (20% coolant oil-in-water).

According to the authors, the control of the grinding temperature makes the workpiece material harder (reduces the softening of workpiece), which contributes to material removal by shear action rather than sliding, thereby improving surface quality. Similar results were observed by Fredj et al. [29] in the grinding of AISI 304 stainless steel with $Al_2O_3$ and the cryogenic cooling (liquid nitrogen) technique. By scanning electron microscope (SEM) images, the authors verified that on the machined surface, cryogenic grinding favored material removal by shearing, which positively affected the surface quality of the ground surface (approximately 40% reduction in Ra parameter in comparison to conventional cutting fluid application).

In this context, considering the surface roughness results observed in this work (Figure 3), it can be inferred that the use of conventional cutting fluid at low temperatures plays a role somewhat similar to that of cryogenic cooling regarding its effect on the surface roughness of the ground surface. By controlling the cutting zone temperature during grinding, the application of cutting fluid at low temperatures reduces adverse thermal effects on the workpiece–wheel interface, including material softening and thermal distortions, which consequently improves surface quality (lower values of roughness Ra). Additionally, it is important to consider that the viscosity of lubricants is in general a temperature-dependent property which increases with temperature drop [39]. Higher viscosity improves lubrication capacity, thereby improving tribological conditions at the cutting zone, which results in the superior finishing of the ground surface (lower values of Ra) [17,40,41].

### 3.2. Images of Machined Surfaces

In Figure 4, the images of ground surfaces of SAE 52100 steel after grinding with the different cutting fluid temperatures used in this work are shown. One notes from Figure 4 that a visible workpiece burn can be observed after grinding with the cutting fluid at room temperature (Figure 4d). Furthermore, it can be observed that the visible marks of the burn are in accordance with the cross-feed value ($a_p$). For the conditions using the cutting fluid at low temperatures (Figure 4a–c), no visible marks of workpiece burn were detected.

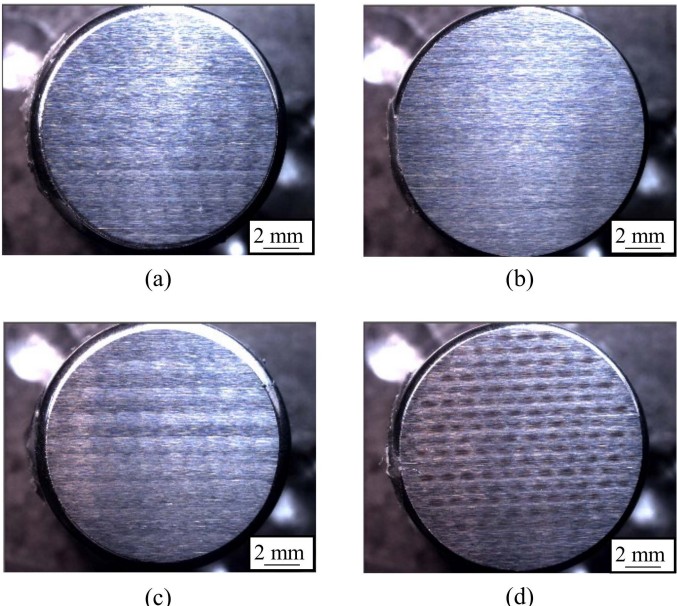

**Figure 4.** Images of the machined surface after grinding with cutting fluid at different temperatures: (**a**) 5 °C; (**b**) 10 °C; (**c**) 15 °C; and (**d**) room temperature.

According to He et al. [42], an oxide film is produced on ground surface in the case of high-temperature conditions at the contact zone, and the degree of thermal damage (e.g., hardness variation) can be associated with the color of the oxide film. Thus, the results presented in Figure 4 suggest that the grinding with the cutting fluid at room temperature

presented a higher hardness variation beneath the machined surface. The surface oxidation is a metallurgical and chemical phenomenon which must be avoided because it leads to a reduction in the component's fatigue life [1]. Tawakoli et al. [43] observed similar burn marks on the ground surface of SAE 52100 hardened steel after grinding with an alumina wheel under different cooling–lubrication conditions. The authors explained that this phenomenon occurs when the specific energy during the process reaches a critical value is related to the cutting conditions (e.g., depth of cut and workspeed).

In this context, it can be inferred from the images of machined surfaces in Figure 4 that the use of the cutting fluid at low temperatures contributed to a reduction in the specific energy during grinding, possibly due to better lubrication capacity (higher viscosity), which results in tangential force reduction [43,44]. Furthermore, since oxidation is favored at higher temperatures [39], lowering the temperature at the cutting zone by using the cutting fluid at low temperatures contributes to minimize the visible burn marks on the ground surface. It is worth mentioning, however, that the absence of visible temper colors on the ground surface does not necessarily mean that no thermal damage occurred.

### 3.3. Microhardness and Microstructure Beneath the Machined Surface

The microhardness of the workpiece after grinding is shown in Figure 5 as a function of depth beneath the ground surface. Each hardness value corresponds to the main average of all six (6) measurements performed for each grinding condition, including the test and replica. The grey dotted line represents the workpiece's microhardness measured prior to the grinding process (885 HV 0.05) and the green area stands for the admissible variation on the workpiece hardness (±2 HRC or ±40 HV). One notes from Figure 5 that all the samples experienced a hardness variation beneath the machined surface, which is evidence that the thermal damage occurred when grinding under the conditions tested in this work.

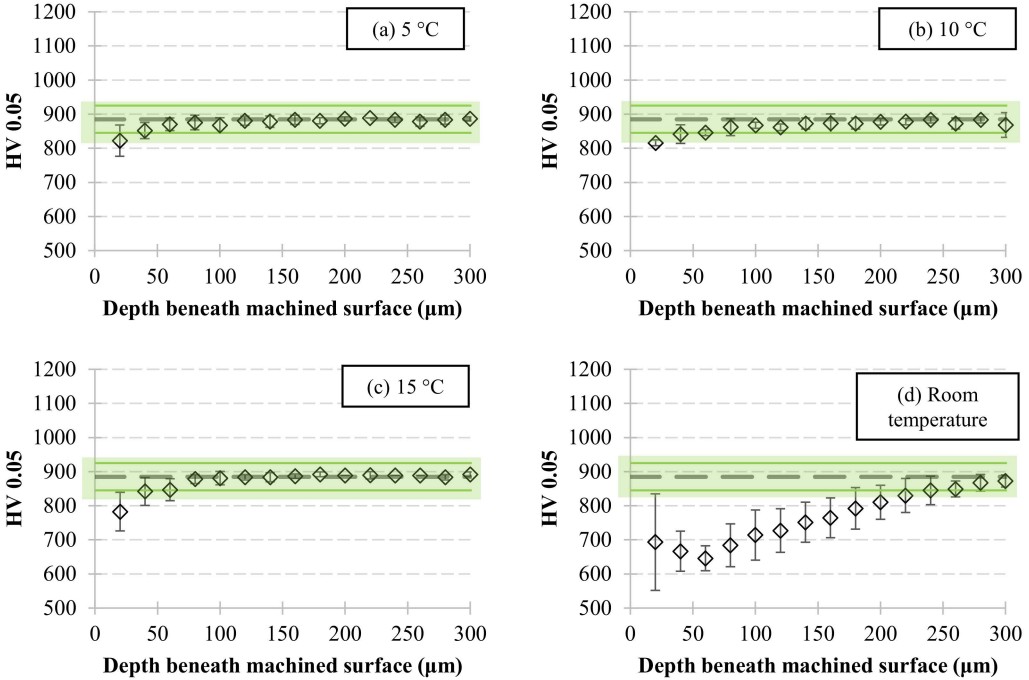

**Figure 5.** Microhardness as a function of depth beneath the machined surface after grinding with the cutting fluid applied at different temperatures: (**a**) 5 °C; (**b**) 10 °C; (**c**) 15 °C; and (**d**) room temperature.

As it can be noticed from Figure 5d, after grinding with the cutting fluid at room temperature (28 ± 1 °C), thermal damage is observed, considering the mean value of hardness up to 240 μm beneath the machined surface. On the other hand, the hardness variation for the conditions with the cutting fluid at low temperatures generated only 20 μm of hardness variation, as shown in Figure 5a–c. Such a result corroborates with the discussion regarding

the images of the machined surface and shows that, considering the conditions used in this work, controlling cutting fluid temperature at low levels (15 °C, 10 °C and 5 °C) was able to reduce the region with hardness variation beneath the ground surface by 92% in comparison to the cutting fluid applied at room temperature (28 ± 1 °C).

Not only was the region with hardness variation reduced with the cutting fluid temperature, but so was the magnitude of the hardness variation. After grinding with the cutting fluid at room temperature, the workpiece presented a mean hardness of 693 HV at 20 μm below the machined surface—which is a 22% reduction compared to the workpiece's hardness prior to grinding. The reduction in the conditions using the cutting fluid at 5 °C, 10 °C, and 15 °C were 7%, 8%, and 11%, respectively.

The thermal damage in the grinding process was a result of the great amount of heat generated at the cutting zone, which is a consequence of rubbing, plastic deformation, and shearing due to the material removal mechanism. As previously mentioned, most of this heat is conducted to the workpiece, especially when using conventional abrasive grinding wheels such as $Al_2O_3$, which contributes to the development of temperatures high enough to cause non-intentional heat treatment near the machined surface.

According to Seidel et al. [45], two different types of thermal damage can be observed: an excessive tempering zone characterized by hardness reduction, and a rehardened zone characterized by the formation of brittle untampered martensite, consequently leading to increased hardness (usually known as a white layer). The tempering phenomenon is associated with carbon diffusion and depends on the temperature and its time of action. In the case of white layer formation, the temperature of the workpiece near the machined surface must be high enough and persist for long enough for reaustenitization to occur, followed by rapid cooling [3].

As it can noticed from Figure 5, all grinding conditions presented a hardness reduction in comparison to the material hardness prior to grinding, and therefore, it is associated with tempering. According to Malkin and Guo [3], this material softening close to the ground surface is common in the grinding of hardened steel, even if no apparent burn marks are visible. Additionally, since thermal damage in grinding is a temperature-dependent phenomenon, the higher the temperature, the higher the thermal damage. In this context, the microhardness analysis of this work suggests that the cutting zone temperature reduces with the cutting fluid temperature, thereby attenuating thermal damage in terms of hardness reduction, as shown in Figure 5.

The relation between the cutting fluid temperature and workpiece thermal damage can be explained by using heat transfer concepts. As previously mentioned, the direct cooling of the cutting zone is one of the most important functions of cutting fluid in grinding [3,10], which occurs through forced convection, which is dependent, among other factors, on the temperature difference between the cutting fluid and the cutting zone. The higher this temperature difference, the higher the heat absorption by the cutting fluid, which decreases the heat partition that is conducted to the workpiece, thereby minimizing thermal damage occurrence. Similar results can be found in the literature regarding cryogenic grinding, as observed by Paul and Chattopadhyay [46].

Lastly, the increase in the cutting fluid viscosity due to the temperature drop must be considered as previously discussed for surface finish results. The higher viscosity also contributes to temperature control by reducing the heat generation by friction at the cutting zone, which corroborates with the surface finish discussion. Thus, the significative reduction in thermal damage when controlling for the cutting fluid temperature at low levels (as shown in Figure 5) is attributed to the combination of better heat dissipation and the higher lubrication capacity of the cutting fluid.

The microstructure beneath the machined surface is shown in Figure 6 for all the cooling–lubrication conditions tested in this work. One notes from Figure 6 that after grinding with cutting fluid at room temperature (Figure 6d), a clearly over-tempered region (dark layer) beneath the ground surface can be observed, indicated by the dashed square. Furthermore, the thickness of this dark layer is approximately 180 μm, which is in good

agreement considering the standard deviation of the microhardness results (Figure 5d). Controlling the cutting fluid temperature at low levels significantly reduced the dark layer thickness, especially at 5 °C and 10 °C, which corroborates the results of microhardness beneath the machined surface.

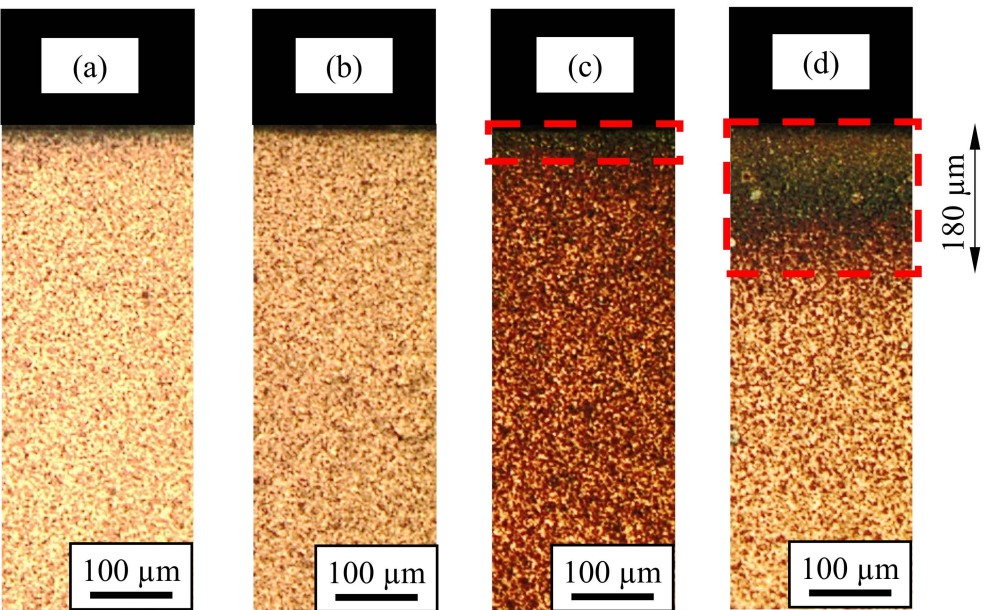

**Figure 6.** Microstructure beneath the machined surface after grinding with cutting fluid at different temperatures: (**a**) 5 °C; (**b**) 10 °C; (**c**) 15 °C; and (**d**) room temperature.

## 4. Conclusions

The grinding of SAE 52100 hardened steel with the $Al_2O_3$ wheel were carried out using a conventional cutting fluid at different temperatures: 5 °C, 10 °C, 15 °C, and room temperature (28 ± 1 °C). The surface integrity of the workpiece (Ra parameter, images of ground surface, and microhardness and microstructure beneath machined surface) was analyzed and the following conclusions can be drawn:

i   Although usually neglected in research papers concerning conventional cutting fluid applications (flood) in grinding, the temperature of the cutting fluid in the instant of its application strongly affected the workpiece surface integrity. Controlling this parameter at low levels (5–15 °C) contributed to the improved surface quality of the workpiece, especially for the lowest cutting fluid temperature (5 °C), which promoted a 31% reduction in Ra roughness in comparison to the cutting fluid at room temperature (28 ± 1 °C).

ii  No visible workpiece burn was observed on the ground surface after grinding with cutting fluid at low temperatures (5–15 °C)—only when grinding with the cutting fluid at room temperature.

iii Controlling the cutting fluid temperature at low levels (5–15 °C) reduced the extension of hardness reduction beneath the round surface to 92% in comparison to the grinding with cutting fluid at room temperature (28 ± 1 °C). Furthermore, the thickness of the over-tempered region (dark layer) beneath the machined surface was significantly reduced by applying the cutting fluid with temperature at low levels.

iv  Controlling the temperature of a conventional cutting fluid at low levels (5–15 °C) is a viable alternative to prevent or attenuate possible thermal damage during grinding.

**Author Contributions:** Conceptualization, R.L.d.P. and R.B.d.S.; methodology, R.L.d.P.; formal analysis, R.L.d.P.; investigation, R.L.d.P.; resources, R.B.d.S.; writing—original draft preparation, R.L.d.P.; writing—review and editing, R.L.d.P., R.d.S.R. and R.B.d.S.; supervision, R.B.d.S.; project administra-

tion, R.B.d.S.; funding acquisition, R.B.d.S. All authors have read and agreed to the published version of the manuscript.

**Funding:** This study was partly funded by the Fundação de Amparo à Pesquisa do Estado de Minas Gerais (FAPEMIG)—grant reference 01/2016 APQ-01119-16 and 002/2018—PPM XII—process n.: PPM-00492-18, the Fundação de Apoio Universitário (FAU)—grant reference 02/2018, Coordenação de Aperfeiçoamento de Pessoal de Nível Superior (CAPES), and the Conselho Nacional de Desenvolvimento Científico e Tecnológico (CNPq)—grant references 140320/2016-4, 426018/2018-4, 310264/2019-7, 141472/2017-0.

**Data Availability Statement:** Not applicable.

**Acknowledgments:** The authors are grateful to the Post Graduate Program of Mechanical Engineering of Federal University of Uberlandia, CAPEX PROEX and the Brazilian foundations Conselho Nacional de Desenvolvimento Científico e Tecnológico (CNPq), Coordenação de Aperfeiçoamento de Pessoal de Nível Superior (CAPES) and the Fundação de Amparo à Pesquisa do Estado de Minas Gerais (FAPEMIG) for their technical and financial support. The authors also thank the Saint-Gobain Abrasives of South America and the Blaser Swisslube by suppling this research with the grinding wheel and the cutting fluid, respectively.

**Conflicts of Interest:** The authors declare no conflict of interest.

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
