# Peer review of "An Approach to Reduce Thermal Damages on Grinding of Bearing Steel by Controlling Cutting Fluid Temperature"

_metals, doi:10.3390/met11101660_

Round 1
Reviewer 1 Report
The reviewer comments of the paper «An Approach to Reduce Thermal Damages on Grinding of Bearing Steel by Controlling Cutting Fluid Temperature»- Reviewer
The authors presented an article «An Approach to Reduce Thermal Damages on Grinding of Bearing Steel by Controlling Cutting Fluid Temperature». The article is interesting and with original research content. Although the article is rather short, it shows the methods and options for solutions and explanations of the results well. However, there are several points in the article that require further explanation.
Comment 1:
The introduction needs to be improved.
Firstly, group quotation is unacceptable in one phrase, for example [16-22]. Break this sentence into parts or individual sentences. For example, ... [...], ... [...], etc. Or one reference - one sentence.
Now the list of references needs to be supplemented with at least 4-6 more references published over the past 5 years. Here are some recent articles:
International Journal of Advanced Manufacturing Technology 2021. doi:10.1007/s00170-021-07785-x
International Journal of Advanced Manufacturing Technology 2017, 91(9-12), 4055–4068. doi:10.1007/s00170-017-0036-4
It is necessary to add a paragraph and a detailed analysis of the studied material of the workpiece. What difficulties are there in the machining and milling process especially? Why is this material so important? Here are just a few articles on 52100 steel:
International Journal of Advanced Manufacturing Technology 2021, 113(11-12), 3329-3342. doi:10.1007/s00170-021-06713-3
International Journal of Advanced Manufacturing Technology 2019, 105(10), 4211-4223. doi:10.1007/s00170-019-04582-5
Comment 3:
- Materials and Methods
Are all figures original? If not needed appropriate citations and permissions.
Add the material chemistry of the SAE 52100 hardened 147 steel stock in a separate table. What is the hardness of the workpiece and how was it measured?
Decipher this designation: what bunch is used, what size of grains, etc. Explain your choice of grinding wheel.
Describe the measurement procedure in more detail. At what point in time? How is the measuring setup set up? How many repetitions of measurements? What statistical methods are used to process experimental results? Describe the experimental stand in more detail. What method of experiment planning is used and why?
Comment 4:
- Results and discussion
Figure 3 is best redrawn in color.
Comment 5:
Conclusions
It is necessary to more clearly show the novelty of the article and the advantages of the proposed method. What is the difference from previous work in this area? Show practical relevance.
Comment 6:
The reference list must be drawn up in accordance with the MDPI requirements.
The article is interesting, but needs to be improved. Authors should carefully study the comments and make improvements to the article step by step. After major changes can an article be considered for publication in the "Metals".
Author Response
RESPONSE LETTER – MANUSCRIPT ID metals-1397792
On behalf of myself and all the authors, I would like to thank you for the considerations and the opportunity given for improving our manuscript. We are also grateful to the reviewers for their valuable comments and suggestions that contributed to improve quality of this manuscript.
The paper was modified accordingly to attend the reviewers’ suggestions and we expect that it is now within the quality demanded by this reputable international journal.
The manuscript has been revised based on reviewers’ comments and suggestions, and detailed corrections and responses are listed below point by point, in blue. The changes on the manuscript are all identified.
Please, do not hesitate in contacting me if further information is required.
Sincerely yours,
Dr. Rosemar Batista da Silva
Federal University of Uberlandia (UFU), School of Mechanical Engineering
Uberlandia, Minas Gerais, Brazil
rosemar.silva@ufu.br

Reviewer 2 Report
Title: An Approach to Reduce Thermal Damages on Grinding of 2
Bearing Steel by Controlling Cutting Fluid Temperature
A very interesting study and specifically one I am aware has not been carried out previously. In short, the temperature effects of cooling of the coolant was investigated in terms of material characteristic outputs such as surface quality and material integrity in terms of unwanted anomalies. Albeit interesting and useful to the grinding community there are still some issues that need clearing up beforehand.
Abstract and Introduction, section and some comments such as grammar and author reference to be taken for the whole document:
The grammar and English need to be reviewed throughout the whole document. Such things such as incorrect use of singular and plural. Also to use ‘the’ in from of nouns as in the case ‘conducted to workpiece’ needs to be changed to ‘conducted to the workpiece.’
During the abstract the word conducted to workpiece appears to be semantically incorrect I would suggest channeled or something similar would be more appropriate.
The first sentence on page 2 needs a reference talking about Al203 as poor heat conductor.
Have you heard of VIPER grinding before? A reference to this in terms of nozzle position is important to reference page 3 third paragraph.
In the introduction there is no mention of CO2 as a coolant – also a consideration for zero-based carbon conditions – this should also be commented on. Does the cooling of coolant also contribute to less CO2?
The sentence needs review first sentence, the final paragraph of section 1.
Also, you mentioned the authors are these the aforementioned or yourselves – you need to be cleared here.
Section 2:
A conventional with aluminum oxide? Please revise this sentence.
Why was 37 m/s used as opposed 40 m/s which is more of an industrial standard?
Dressing speed of 140 mm/min suffices more than equals to
Third paragraph “After process” à which process – please be specific
Usually, temperatures would be displayed ascending rather than descending – see Table 1 and throughout the paper.
Section 3:
The first sentence needs revision.
How many surface measurements were carried out ? I.e. what was the repeatability of the tests?
Any reason for the change of slope from CF at 15 degrees C to CG at 10 degrees C and then back to CF at 5 degrees C.
Careful again with the article, for example ‘as cutting fluid’ should be ‘as a cutting fluid.’ There are more errors similar to this – please revise throughout.
The last sentence before section 3.2 need revision and be written more accurately.
Careful of spelling colling should be cooling – check spelling through the manuscript.
The penultimate paragraph and first sentence need revision.
Conclusions:
Point (ii) needs revision
Author Response
Please, see attachment.
The manuscript has been revised based on reviewers’ comments and suggestions, and detailed corrections and responses are listed below point by point, in blue. The changes on the manuscript are all identified.
Best regards.

Reviewer 3 Report
The work is very good. Please consider these following minor suggestions:
- Abstract is too long. Please include only important details of paper.
- In introduction, other studies on different materials could be included to strengthen the literature part. Please follow papers of Prof. Grzegorz, Munish Kumar Gupta, Prof Murat Sarikaya,
- I think its better to show real experiment picture.
- Results and discussion also need more technical details.
Author Response
Please see attachment
Dear Reviewer 3
RESPONSE LETTER – MANUSCRIPT ID metals-1397792
On behalf of myself and all the authors, I would like to thank you for the considerations and the opportunity given for improving our manuscript. We are also grateful to the reviewers for their valuable comments and suggestions that contributed to improve quality of this manuscript.
The manuscript has been revised based on reviewers’ comments and suggestions, and detailed corrections and responses are listed below point by point, in blue. The changes on the manuscript are all identified.
Best regards.
Rosemar

Round 2
Reviewer 1 Report
The authors have improved the article according to the comments. However, before publication, it is important that in Table 1 the authors replace "," with "." in accordance with the rules of the English language.
Author Response
Uberlandia, Minas Gerais, Brazil, October 8, 2021.
To: Editorial board of Proceedings of Metals
From: Dr Rosemar Batista da Silva
RESPONSE LETTER – MANUSCRIPT ID metals-1397792 – R2
On behalf of myself and all the authors, I would like to thank you for the consideration and the opportunity given for improving our manuscript again. We are also grateful to the reviewer for the meticulous reading.
The manuscript has been revised based on reviewer’s comment. The response is listed below, in blue. The changes on the manuscript are all identified.
#REVIEWER 1:
The authors have improved the article according to the comments. However, before publication, it is important that in Table 1 the authors replace "," with "." in accordance with the rules of the English language.
Thank you for your comment. Table 1 was corrected using the decimal point.
Therefore, the paper was modified accordingly to attend the reviewer’s suggestion and we expect that it is now within the quality demanded by this reputable international journal.
Please, do not hesitate in contacting me if further information is required.
Sincerely yours,
Dr. Rosemar Batista da Silva
Federal University of Uberlandia (UFU), School of Mechanical Engineering
Uberlandia, Minas Gerais, Brazil
rosemar.silva@ufu.br